# Two Approaches to Triple Antithrombotic Therapy in Patients with Acute Coronary Syndrome and Non-Valvular Atrial Fibrillation Treated with Percutaneous Coronary Intervention: Which Is More Efficient and Safer?

**DOI:** 10.3390/diagnostics13193055

**Published:** 2023-09-26

**Authors:** Witold Bachorski, Jakub Bychowski, Marcin Gruchała, Miłosz Jaguszewski

**Affiliations:** Department of Cardiology, Medical University of Gdansk, M. Skłodowskiej-Curie 3a, 80-210 Gdansk, Poland

**Keywords:** NOAC, VKA, ACS, AF, PCI

## Abstract

Introduction: Patients with acute coronary syndrome (ACS) and atrial fibrillation (AF) treated with percutaneous coronary intervention (PCI) are at high risk of bleeding and thromboembolic events. Thus, optimal treatment strategies in this challenging subset have been controversial. Herein, we aim to investigate different triple antithrombotic treatment (TAT) strategies in patients with ACS and AF after PCI. Methods: This was a retrospective, single-center study based on all consecutive patients with the diagnosis of ACS and AF treated with vitamin K antagonists (VKA) or non-vitamin K antagonist oral anticoagulants (NOAC) plus dual antiplatelet therapy using a P2Y12 inhibitor (clopidogrel) and aspirin (for 1 to 3 months) and observed for 12 months for major adverse cardiac events (MACE) and major or clinically relevant non-major bleeding incidents. Results: MACE occurred in 26.6% of patients treated with the VKA and 30.9% with NOAC (*p* = 0.659). Bleeding occurred in 7.8% of patients treated with VKA and 7.4% with NOAC (ns). Conclusions: Among patients with ACS and AF who had undergone PCI, there was no significant difference in the risk of bleeding and ischemic events among those who received TAT with NOAC and VKA.

## 1. Introduction

Coronary artery disease is a condition of origin in atherosclerotic stenoses of coronary arteries. It is a chronic condition that consists of episodes of relative stability with no symptoms or their slow progression and moments of exacerbation of ailments and rapid acceleration of symptoms, namely acute coronary syndrome (ACS). Percutaneous coronary intervention (PCI) with implantation of a drug-eluting stent (DES) and dual antiplatelet therapy (DAPT) is a recommended standard of care. The advancement in the technology of coronary angioplasty, from the use of balloons through bare metal stents (BMS) to DES, is an ongoing process. There is scientific evidence that the newest generation of ultrathin strut DES improve patients’ outcome compared to older-generation stents [1]. These stents are being used more and more frequently in daily clinical practice. The use of stents dramatically decreased the incidence of restenosis after balloon angioplasty, although the problem of in-stent thrombosis emerged. The regimens based on aspirin, heparin, or VKA were introduced. However, after the development and advancement of P2Y12 inhibitors, treatment strategies were changed to antiplatelet therapy. DAPT improved efficacy and safety compared to aspirin and VKA [2]. Acetylsalicylic acid (ASA) already has a well-established position in primary and secondary prevention of cardiac events. Large clinical trials proved that using ticagrelor or prasugrel as an addition to ASA results in an immediate and continuous antiplatelet effect and reduction of ischemic events rate [3]. It is extremely important in the course of ACS, which is a diffused inflammatory process in the coronary arteries associated with hypercoagulability.

What is more, atrial fibrillation (AF), the most common arrhythmia worldwide, increases the risk of thromboembolic events [4]. It is believed that more than 30% of patients suffer from AF and CAD simultaneously, and approximately 20% of patients with AF need PCI due to CAD [5]. Vitamin K antagonists (VKA) or non-vitamin K antagonist oral anticoagulants (NOAC) are the standard of care for stroke prophylaxis. On the other hand, CAD raises the CHA_2_DS_2_-VASc score, hence increasing the risk of stroke. Therapy with aspirin and P2Y12 receptor inhibitors (ticagrelor, clopidogrel, and prasugrel) DAPT is a gold standard for secondary prevention of myocardial infarction (MI). Thus, triple antithrombotic therapy (TAT) with VKA or NOAC plus two antiplatelet agents is the standard of care after PCI or ACS for patients with AF. Still, both strategies are associated with a high risk of bleeding [6]. Therefore, the length of antiplatelet therapy as an addition to anticoagulant should be set based on the patient’s diagnosis, type of treatment, and comorbidities.

There are ongoing trials and analyses comparing both (VKA and NOAC) types of treatment in terms of ischemic and bleeding events. Taking into account various meta-analyses that summarized results of trials such as PIONEER AF-PCI, RE-DUAL PCI, AUGUSTUS, and ENTRUST-AF PCI, it is believed that TAT with the use of NOAC reduces major bleeding events rate compared to VKA [7,8]. However, those studies analyzed patients undergoing PCI electively and due to ACS as a whole group. In ACS, the treated lesion heals slower than in CCS. Those patients are at higher risk of thromboembolic events. Therefore, herewith, we sought to investigate the safety and efficacy of TAT using either NOAC or VKA in the real-life population of patients who underwent PCI due to ACS only.

## 2. Methods

### 2.1. Subset

This analysis of a retrospective, single-center observation from the Acute Coronary Syndrome Registry is based on the medical records of all consecutive patients treated in the University Clinical Centre of the Medical University of Gdańsk. All patients who were at least 18 years of age, had nonvalvular AF, and had successfully undergone PCI with a DES within the previous 120 h were treated with DAPT (aspirin + clopidogrel) with the addition of NOAC or VKA. The type of treatment was a mutual agreement between the patient and the physician based on the patient’s socioeconomic status and comorbidities. The selection of a substance within a drug group was a physician’s choice. The duration of treatment was designed according to the current knowledge and guidelines. The authors vouch for the accuracy of the data and analyses, and an appropriate bioethics committee is aware of this registry.

The Acute Coronary Syndrome Registry consisted of 1095 patients admitted to the Clinic of Cardiology from 1 January 2015 to 31 December 2016. All patients were diagnosed with MI or unstable angina (UA). Patients treated conservatively were excluded from the analysis. The authors of this observation analyzed only the patients with a new type of DES implanted. Therefore, patients after plain old balloon angioplasty and BMS implantation were excluded from the analysis. The authors of this study gathered information on demography, comorbidities, medical history, medications at admission and discharge, symptoms, laboratory results during hospitalization, echocardiography results, and detailed information on coronary angiography and PCI, including access site, assessment of significant stenoses, intravascular imaging, type of lesion (de novo, restenosis, or thrombosis), stent model and size, maximum pressure used for dilatation, and effect of the procedure. Patients were observed within 12 months for death from any cause, bleeding incidents, MACE including death, stroke, unplanned PCI, coronary artery bypass grafting (CABG), and MI. The gathering of follow-up data was designed based on the ICD-10 and ICD-9 codes used for the standard coding of the diagnosis and procedures. The proper analyses were designed, the National Health Fund analyzed data, and results were given to the authors of this study. Observation of the events ceased on the first incident. Major bleeding was defined according to the International Society on Thrombosis and Haemostasis (ISTH) Bleeding Definitions [9]. Other bleeding incidents were categorized as minor bleeding.

### 2.2. Statistical Analysis

Continuous data are presented as mean ± SD or median and compared with the Student *t*-test or Mann–Whitney. Categorical variables are presented as counts and compared using the χ^2^ test. For all endpoints, Kaplan–Meier estimates were calculated and presented graphically. *p* < 0.05 indicates statistical significance. The statistical analysis was performed with the use of Statistica 13.

## 3. Results

The baseline characteristics of the patients are shown in Table 1. A total of 64 patients were discharged on TAT with VKA and 81 on TAT with NOAC (for the assignment of patients into the study group, see Figure 1). Both groups had no statistically significant differences regarding comorbidities or atherosclerosis risk factors. However, there was a visible trend of fewer past smokers in the NOAC vs. VKA group (48.4% vs. 34.2%, *p* = 0.09) and higher creatinine clearance in the NOAC vs. VKA group (73.8 ± 35.9 vs. 86.1 ± 38.9; *p* = 0.07). Patients discharged on NOAC vs. VKA had better ejection fraction (50.9% ± 19 vs. 40.6% ± 12.8; *p* = 0.0007). Details regarding the treatment method and final diagnosis are shown in Table 1.

MACE occurred in 17 of 64 (26.6%) patients treated with VKA and 25 of 81 (30.9%) patients treated with NOAC (*p* = 0.659, see Figure 2). The incidence of death by any cause (4.7% vs. 9.9%; *p* = 0.378, see Appendix A), stroke (1.6% vs. 2. 5%; *p* = ns, see Appendix A), myocardial infarction (15.6% vs. 14.8%; *p* = ns, see Appendix A), CABG (0 vs. 2.5%; *p* = 0.576, see Appendix A), and rePCI (9.4% vs. 7.4%; *p* = 0.924, see Appendix A) did not differ significantly between both groups.

Any bleeding occurred in 5 of 64 (7.8%) patients treated with VKA and 6 of 81 (7.4%) patients treated with NOAC (*p* = ns, see Figure 2). No differences were recorded in minor bleeding (6.3% vs. 4.9%; *p* = ns, see Appendix A) or major bleeding (7.8% vs. 7.4%; *p* = ns, see Appendix A).

## 4. Discussion

Our study summarises the observation on using VKA or NOAC within TAT after the diagnosis and invasive treatment of ACS. We analyzed the data in terms of efficacy and safety. As a result, interestingly, there is no significant difference in the 12-month follow-up in both ischemic and bleeding endpoints. Patients who used NOACs were numerically less likely to have rePCI or MI. On the other hand, patients on VKA were reported with fewer deaths, although no significant differences were achieved. There was more MACE in the group on NOAC due to deaths, CABG, and stroke. However, the number of CABG and stroke was too limited for precise calculations. There was no statistical difference in minor, major, and overall bleeding.

Medications that inhibit factor Xa (apixaban, rivaroxaban, and edoxaban) and thrombin (dabigatran) are recommended for patients with AF due to a lower number of strokes, mortality, or intracranial bleeding compared to the therapy of VKA [10]. Although the strategy with VKAs has been recently disregarded in patients with AF when stent implantation has to be implemented in the acute course of ACS, we decided to compare two different TAT strategies.

The meta-analysis by Eyileten et al. found that the general effect of TAT with NOAC vs. VKA is the reduction of bleeding events with no increase in ischemic events [7]. This observational work has analyzed the results of major trials such as PIONEER AF-PCI (rivaroxaban + DAPT vs. VKA + DAPT after PCI), RE-DUAL PCI (dabigatran + P2Y12 inhibitor vs. warfarin + DAPT after PCI), AUGUSTUS (apixaban or VKA + P2Y12 inhibitor with or without ASA after ACS or PCI), and ENTRUST-AF PCI (edoxaban + clopidogrel vs. VKA + DAPT after PCI) in which the authors analyzed the results of both elective and urgent patients undergoing a PCI. Therefore, our work’s less promising NOAC results could be due to the group of patients limited to ACS. It is widely known that those patients have a higher risk of thrombotic events, especially during the 12 months after the acute incident. The comparison of the treatment of edoxaban + clopidogrel vs. VKA + DAPT (ENTRUST-AF PCI) was similar to our results; there were no differences between both groups in the number of bleeding events in the 12 months of observation [11]. On the other hand, the analysis of the AUGUSTUS trial by Alexander et al. showed promising results for the strategies with NOACs. They observed patients for 30 days and 30 days to 6 months after the ACS and/or PCI. There was less major bleeding and hospitalizations within 30 days by 2.4% and in 30 days to 6 months by 1.8%. Moreover, ischemic events and hospitalizations were less frequent by 1.61% in 30 days of observation and by 1.16% in 30 days to 6 months [12]. However, Lopes et al., in an analysis of WOEST (NOAC or VKA + clopidogrel alone or DAPT after PCI), PIONEER AF-PCI, RE-DUAL PCI, and AUGUSTUS [8] compared bleeding and efficacy of various TAT and DAPT. The authors showed comparable results between different antithrombotic treatment methods regarding MACE, with no significant differences for MI, stroke, or stent thrombosis. In comparison between VKA and DAPT or NOAC and DAPT, the treatment strategy of NOAC plus DAPT was safer regarding bleeding but without statistical significance. There were fewer bleeding events, including TIMI major, both TIMI major and minor or intracranial hemorrhage. Our observation showed a lower percentage of bleeding events on NOAC. Still, the small number of bleeding events does not allow us to conclude with a clear statement regarding the beneficial effect of this strategy. Similar conclusions are in the analysis by Kuno et al. [13]. There were no significant differences in ischemic events. The highest bleeding risk was in the group with VKA and DAPT. Notably, in the sub-analysis of the ENTRUST-AF, ACS patients were observed compared to chronic coronary syndrome (CCS) patients with AF. As for the ACS, the primary bleeding endpoint occurred in 15.2% of NOAC patients vs. 20.3% of VKA patients (*p* = 0.063). MACE occurred in 8.5% vs. 7.2% (*p* = 0.53), respectively. The net clinical benefit was only slightly towards NOAC (*p* = 0.94) [14]. So far, all RCTs showing better results after TAT with NOAC vs. TAT with VKA included mixed subsets comprising patients with CCS and ACS treated with PCI or conservatively. Thus, such observations comparing two different TAT strategies within the ACS registry, i.e., clinical practice subset with no exclusion criteria, are still strongly needed. A study similar to GEMINI-ACS or APPRAISE-2 is needed. The first study compared the use of rixaroxaban 2.5 mg bidaily + clopidogrel to ASA + clopidogrel or ticagrelor after ACS. As a result of this analysis there were no significant differences in either bleeding or ischemic events [15]. The latter study compared patients after recent ACS (however, less than half of whom underwent PCI) who were randomized to two groups: apixaban 5 mg bidaily + DAPT or placebo + DAPT. The trial was terminated due to an increased number of major bleeding in the group with apixaban. However, the increase in bleeding was not associated with the significant decrease in ischemic events. What is more, a higher number of fatal and intracranial bleeding occurred in the group who received apixaban. Patients who underwent PCI had the same results as patients treated with a different approach. Although these studies analyzed the population of patients with ACS, they did not take into consideration all the factors, such as AF as comorbidity or VKA treatment. Therefore, the groups that were observed are only a part of the whole population of patients from a daily clinical routine [16]. Although treatment with the use of NOAC is favored, we have to remember that VKA treatment is still popular in clinical practice mainly due to economic factors concerning a significant difference in price compared to NOAC, which is important for many patients. According to the RIVA-PCI study, which analyzed current antithrombotic patterns in the treatment of patients with AF and PCI, TAT is given to 26%, NOAC is used in 82.4%, and VKA is still used in 11.5% of patients [17].

Our study should be interpreted in the context of its limitations. This was an observation from a single, high-volume center ACS registry. However, all patients were treated with TAT, and the population was limited to the ACS with no exclusion criteria (except for the old methods of invasive treatment). The follow-up data were ascertained with a National Health Fund clinical outcomes database, where all patients were linked to the MACE records. Secondly, although ascertainment of the clinical outcomes using a central national database has already been described as accurate as prospective data collection, we could not fully assess the in-stent thrombosis and restenosis events due to the limitations of the National Health Fund data. This is a retrospective study, with all the limitations that this implies.

## 5. Conclusions

In patients with ACS and AF undergoing PCI, there was no statistical significance in MACE and bleeding events based on the triple anticoagulation therapy with NOAC vs. VKA in 12 months of follow-up.

## Figures and Tables

**Figure 1 diagnostics-13-03055-f001:**
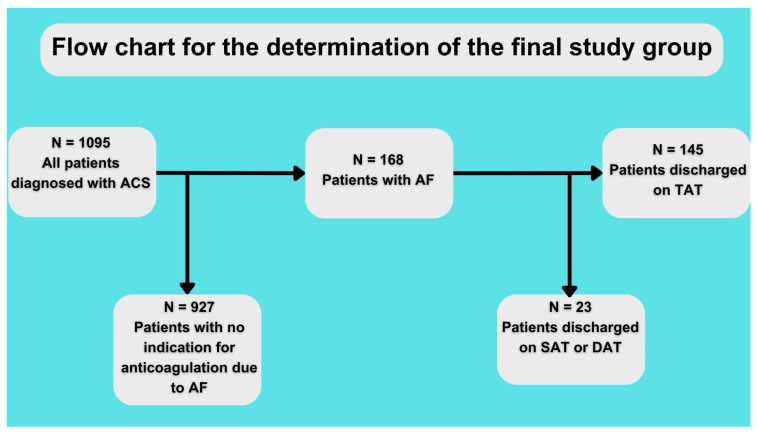
Flow chart for study. ACS—acute coronary syndrome, AF—atrial fibrillation, SAT—single antithrombotic therapy, DAT—double antithrombotic therapy, and TAT—triple antithrombotic therapy.

**Figure 2 diagnostics-13-03055-f002:**
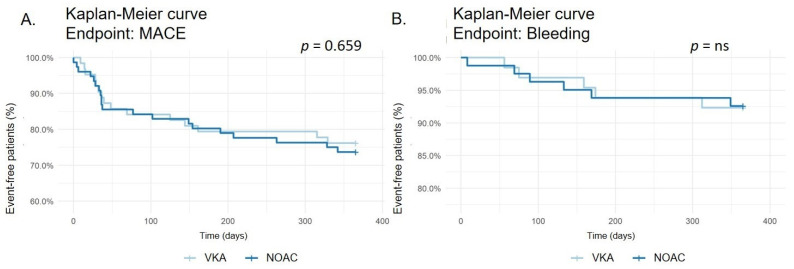
Kaplan–Meier survival curve for major adverse cardiac events (MACE) (**A**). Kaplan–Meier survival curve for bleeding events (**B**).

**Table 1 diagnostics-13-03055-t001:** Baseline characteristics and final diagnosis.

Characteristic	VKA (*n* = 64)	NOAC (*n* = 81)	*p* Value
Age, y	70.3 ± 10.8	71.8 ± 13.5	0.18
Women, *n* (%)	19 (29.7)	29 (35.8)	0.43
Length of hospitalization, d	9.3 ± 4.8	8.6 ± 5.8	0.14
BMI, kg/m^2^	28.7 ± 5.2	29.6 ± 5.7	0.37
Arterial hypertension, *n* (%)	51 (79.7)	67 (82.7)	0.64
Diabetes, *n* (%)	20 (31.2)	29 (35.8)	0.56
Hypercholesterolemia, *n* (%)	52 (81.2)	66 (81.5)	0.97
Smoking, *n* (%)	13 (20.6)	16 (20)	0.92
Smoking history, *n* (%)	30 (48.4)	27 (34.2)	0.09
Creatinine clearance	73.8 ± 35.9	86.1 ± 38.9	0.07
Prior myocardial infarction, *n* (%)	27 (42.2)	34 (42)	0.97
Prior percutaneous coronary intervention, *n* (%)	20 (31.3)	32 (40.5)	0.25
Prior coronary artery bypass grafting, *n* (%)	11 (17.2)	13 (16.3)	0.88
Chronic obstructive pulmonary disease, *n* (%)	4 (6.3)	4 (5)	0.73
Ejection fraction, %	40.6 ± 12.8	50.9 ± 19	0.0007
Systolic blood pressure, mmHg	137.6 ± 22.1	140.2 ± 25	0.5
Heart rate, bpm	88.7 ± 26.8	90.7 ± 30.2	0.7
Diagnosis			
STEMI, *n* (%)	13 (20.3)	18 (22.2)	0.78
NSTEMI, *n* (%)	34 (53)	41 (50.6)	0.76
UA, *n* (%)	17 (26.6)	22 (27.2)	0.94
EuroScore 2	7.9 ± 6.9	8.2 ± 5.9	0.55

STEMI—ST segment elevation myocardial infarct, NSTEMI—non-ST segment elevation myocardial infarct, UA—unstable angina.

## Data Availability

Restrictions apply to the availability of these data. Data were obtained from the National Health Fund and are not available due to legal reasons.

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
