# Peer review of "Two Approaches to Triple Antithrombotic Therapy in Patients with Acute Coronary Syndrome and Non-Valvular Atrial Fibrillation Treated with Percutaneous Coronary Intervention: Which Is More Efficient and Safer?"

_diagnostics, 2023, doi:10.3390/diagnostics13193055_

Round 1

Reviewer 1 Report

Comments and Suggestions for Authors

Present retrospective manuscript has nicely demonstrated that on retrospective single center setup the use of anticoagulation (both NOAK and VKA) parallel to TAT is relatively safe for PCI patients (acute coronary syndrome) with FA.

Some comments should be addressed. Title would be more tempting if revised. 

It would be essential to have a flow chart for the study. Over 1000 patients in the registry, but how and on what criteria the final sample was achieved? A figure with chart might provide clear and easy to interpret way?

It appears that authors have only copy pasted the template for the role of authors etc in the last part of the manuscript. This should be corrected.

Author Response

Thank you for your remarks.

The title of our manuscript is descriptive. However, we have edited it in hope to make it more interesting.

The flow chart for the study was included as Figure 1.

Template for the role of authors was not included by authors of this manuscript. It was probably added by editors. However, we have completed it.

Reviewer 2 Report

Comments and Suggestions for Authors

This paper represents very interesting and actual topic, triple antithrombotic therapy among patients who had stent implantation due to acute coronary syndrome and atrial fibrillation.

The paper is generally well written; the methodology is very good.

However, I have some comments.

1. In the Methods section, subsection Statistical analysis, the sentence “The statistical analysis was performed with use of Statistica 13” should be moved to the end of the paragraph.

2. It was not defined what leads doctors to choose VKA or NOAK as anticoagulant drug. Was it doctor’s own decision or some other things like social status, presence of comorbidities, kidney function...

3. It would be interesting if the authors would present types of P2Y12 inhibitors that were used among NOAC and VKA group of patients (Clopidogrel, Ticagrelor, Prasugrel) in Table 1.

Author Response

Thank you for your suggestions.

Ad. 1 The sentence was moved as suggested.
Ad. 2 We have added an explanation to the decision of choice between VKA and NOAC. It was based on patients financial status and comorbidities.
Ad. 3 The registry is based on hospitalizations in 2015 and 2016. Ticagrelor and prasugrel was not commonly used in Poland due to the availability and price. All patients were given clopidogrel. We have added a note to the manuscript.

Reviewer 3 Report

Comments and Suggestions for Authors

The research is a brief report on "Safety and Efficacy of Non-Vitamin K Antagonist Oral  Anticoagulants Vs Vitamin K Antagonists in Triple3 Antithrombotic Therapy in Patients with Acute Coronary 4Syndrome and Non-Valvular Atrial Fibrillation Treated with  Percutaneous Coronary Intervention"

The paper is of interest  however the comparison of the two groups is not possible without the indication of time in the therapeutic range of patients in VKA 

Comments on the Quality of English Language

Minor typos

Author Response

We agree with this remark. The incompleteness of data is one of the limitations of a retrospective study. We have added a sentence that the study is retrospective and it is a limitation of our work.

Reviewer 4 Report

Comments and Suggestions for Authors The article titled "Safety and Efficacy of Non-Vitamin K Oral 2-Antagonists Anticoagulants versus vitamin K antagonists in Triple 3 Antithrombotic therapy in patients with acute coronary disease 4 Syndrome and nonvalvular atrial fibrillation treated with 5 Percutaneous Coronary Intervention" is very important given the importance of anticoagulation with dual antiplatelet therapy. As a suggestion, place a bibliography of 2023 Within the limitations of the work, it should be considered that the study is retrospective, with the limitations that this implies.

Author Response

Thank you for your remarks.

We have added a suggested note. Data on this topic is scarce in publications in 2023. However, we have added an interesting bibliography position.

Round 2

Reviewer 1 Report

Comments and Suggestions for Authors

Authors have made all required correction during revision. No further comments.

Reviewer 3 Report

Comments and Suggestions for Authors

No other issue